Association of ApoE gene polymorphisms with serum lipid levels and the risk of type 2 diabetes mellitus in the Chinese Han population of central China

Zeng Yali
Wen Shuang
Huan Lijun
Xiong Liang
Zhong Botao 2005ly0915@hust.edu.cn
Wang Pengyun wpy0110@mail.hust.edu.cn
Department of Clinical Laboratory, Liyuan Hospital, Tongji Medical College, Huazhong University of Science and Technology , Wuhan , China
Gould Gwyn
Electronic publication date: 2023 Apr 24
Publication date: 2023
Volume: 11
Electronic Location ID: e15226
Received 2022 Dec 29; Accepted 2023 Mar 23
Copyright: ©2023 Zeng et al.
Copyright year: 2023
Copyright holder: Zeng et al.
License: This is an open access article distributed under the terms of the Creative Commons Attribution License, which permits unrestricted use, distribution, reproduction and adaptation in any medium and for any purpose provided that it is properly attributed. For attribution, the original author(s), title, publication source (PeerJ) and either DOI or URL of the article must be cited.
License URL: https://creativecommons.org/licenses/by/4.0/

Keywords: Apolipoprotein E, Type 2 Diabetes mellitus, Polymorphisms, Lipids

Funding: The China National Natural Science Foundation 82070355 This work was funded by the China National Natural Science Foundation, grants number: 82070355. The funders had no role in study design, data collection and analysis, decision to publish, or preparation of the manuscript.

==============================
Background

Apolipoprotein E (ApoE) is involved in lipid transformation and metabolism. Although some studies have examined the association between ApoE polymorphisms and the risk of type 2 diabetes mellitus (T2DM), the findings differ depending on the location and population.

Methods

A total of 1,738 participants, including 743 patients with T2DM and 995 controls without T2DM, were enrolled from central China, and ApoE polymorphisms, 388T > C (rs429358) and 526C > T (rs7412), were genotyped. The association between ApoE alleles and T2DM and blood lipid levels was analyzed. Logistic regression analysis was performed to evaluate the interactions between ApoE polymorphisms and various factors, such as age, sex, and prevalence of hypertension in patients with T2DM.

Results

The genotype ɛ3/ɛ4 and ɛ4 alleles of ApoE were associated with T2DM risk in the Chinese Han population in central China. Moreover, in patients with T2DM, participants in the E4 (ɛ3/ɛ4, ɛ4/ɛ4) group had significantly higher lipid profiles than those in the E3 (ɛ3/ɛ3) group, whereas participants in the E2 group (ɛ2/ɛ2, ɛ2/ɛ3) showed lower total cholesterol, low-density lipoprotein cholesterol, and ApoE-A1 levels than those in the E3 (ɛ3/ɛ3) group. The results from the current study may help in understanding ApoE polymorphisms and lipid profiles in the Chinese Han population.

Introduction

Type 2 diabetes mellitus (T2DM) is a metabolic disorder characterized by chronic insulin resistance and persistent hyperglycemia and is a growing health problem worldwide (Ahmad et al., 2022). According to Global Burden of Disease data, 460 million people worldwide were affected by T2DM in 2019, with an age-standardized prevalence of 6.0% in men and 5.0% in women (Khan et al., 2020; Tinajero & Malik, 2021). In 2019, it was estimated that 116.4 million adults in China had T2DM (Saeedi et al., 2019; Jia et al., 2019). T2DM is a substantial independent risk factor for cardiovascular diseases (CVDs), such as coronary artery disease (CAD), myocardial infarction, peripheral vascular disease, stroke, and heart failure (Strain & Paldanius, 2018; Yun & Ko, 2021). The long-term effects of T2DM can lead to morbidity and mortality (Deshpande, Harris-Hayes & Schootman, 2008). People with diabetes have a two- to four-fold higher risk of dying from CVD than those without diabetes; over 65% of deaths in people with diabetes are due to CVDs, of which T2DM predominates (Lloyd-Jones et al., 2009).

Numerous risk factors have been recognized to affect the complex root causes of T2DM. Changes in living conditions and lifestyle have largely been attributed to the rise in T2DM cases. These changes have decreased the nutritional value of food and increased sedentary behavior, which has resulted in overweight and obesity in areas that were previously disproportionately affected by undernutrition (Kolb & Martin, 2017). It is widely acknowledged that lifestyle, environmental, and genetic variables play a role in the development of T2DM (Toi et al., 2020). The etiology and development of T2DM are influenced by intricate interactions between various genes and several environmental variables (Geng & Huang, 2020). The development of efficient preventative methods to lower the ever-rising incidence of T2DM, as well as the efficacy and accuracy of treatment and prevention strategies, can be made easier by a greater understanding of the role of genetic variables in the etiology of the disease.

Apolipoprotein E (ApoE), which is mainly synthesized by the liver, can bind to chylomicron, high-density lipoprotein cholesterol (HDL-C), low-density lipoprotein cholesterol (LDL-C), and very low-density lipoprotein-cholesterol (VLDL-C) and cause the transformation and metabolism of lipids (Marais, 2019a). Human ApoE is located on chromosome 19q13.2, and its two common single nucleotide polymorphisms (SNPs), 388T >C (rs429358) and 526C >T (rs7412), can form three haplotypes (ɛ2 (388 T–526 T), ɛ3 (388 T-526C), ɛ4 (388C-526C)) and six genotypes (ɛ2/ɛ2, ɛ2/ɛ3, ɛ2/ɛ4, ɛ3/ɛ3, ɛ3/ɛ4, ɛ4/ɛ4) (Seripa et al., 2011). ApoE polymorphisms are reportedly associated with the transcription level of ApoE, serum total cholesterol (TC), triglyceride (TG), HDL, LDL, and VLDL levels, as well as the risk of dyslipidemia (Khalil et al., 2021). ApoE alleles, especially ɛ4, are among the candidate risk factors that are most likely associated with the risk of CAD in patients with T2DM, and rs157582 in the TOMM40-ApoE region shows a significant association with T2DM (p = 2.8×10−9) (Gao et al., 2021a; Gao et al., 2021b; Larifla et al., 2017; Cook & Morris, 2016a). Although some studies have examined the association between ApoE polymorphisms and the risk of T2DM, the findings differ depending on the location and population. In this study, ApoE polymorphisms and T2DM, as well as their effects on lipid profiles, were examined in the central Chinese Han population.

Materials & Methods

Study participants

Between January 2019 and May 2022, a total of 1,738 individuals from Liyuan Hospital, Tongji Medical College, Huazhong University of Science and Technology, comprising 743 patients with T2DM and 995 controls without T2DM, were registered All participants were unrelated members of the Han Chinese community in central China. With a plasma glucose level >200 mg/dL (11.1 mmol/L), a fasting plasma glucose concentration >126 mg/dL (7.0 mmol/L) after at least 8 h, or a 2-h plasma glucose level >200 mg/dL (11.1 mmol/L) during an oral glucose tolerance test, T2DM was identified as the presence of diabetes (OGTT) (Jia et al., 2019). Each glucose test was repeated the following day to confirm the diagnosis. Patients with acute complications of diabetes and diseases affecting metabolism, such as thyroid or parathyroid glands, were excluded.

The control group consisted of healthy individuals who had undergone a physical examination, had fasting blood glucose levels <126 mg/dL (7.0 mmol/L), had no history of diabetes, and were not currently taking any medication to lower their glucose levels (Cheng et al., 2011).

The study protocol was approved by the Huazhong University of Science and Technology (HUST) Ethics Committee on Human Subject Research and Liyuan Hospital Affiliated to Tongji Medical College of HUST (HUST [2020]IEC(SQ05)). The study also complies with the principles of the Declaration of Helsinki. All the participants provided written informed consent.

Plasma lipid measurements

Peripheral blood samples (two mL each) were collected from all individuals, which were then centrifuged at 4,200 g for 10 min to separate the supernatant plasma samples. A Beckman AU5800 automatic biochemical analyzer was used to assess the plasma concentrations of TG, TC, HDL-c, LDL-c, Apolipoprotein A1 (Apo-A1), and Apolipoprotein B (Apo-B).

Genomic DNA extraction and genotyping

Approximately 2 ml of peripheral blood samples was used to prepare genomic DNA using TIANGEN® Genomic DNA purification kit (Beijing, China) according to a previous study (Tan et al., 2021).

Genotyping of ApoE polymorphisms was performed using a PCR fluorescence probe detection kit (Youzhiyou, Wuhan, Hubei, China), which consisted of 0.2 µL (25 ng) genomic DNA, 23 µL reaction solution, including PCR buffer, dNTPs, specific primers and probes, and internal standard primers (Youzhiyou, Wuhan, Hubei, China). The reaction mixture was initially denatured at 37 °C for 10 min, pre-denatured at 95 °C for 5 min, followed by 40 cycles at 95 °C for 15 s and 62 °C for 30 s. The primers used to amplify exon 4 of ApoE containing rs429358 and rs7412 were 5′ -GCACGGCTGTCCAAGGAGCTGCA-3′ (forward) and 5′-CGCACGCGGCCCTGTTCCACC-3′ (reverse). The fluorescence signals were collected as FAM (ApoE2 526C, ApoE4 388T), VIC (ApoE2 526T, ApoE4 388C), and ROX (internal standard). The sequences of the probes were found as follows: rs429358: FAM-CGGCCGCACACGTCCT-TAMRA, VIC- CGGCCGCGCACGTCCT-TAMRA, and for rs7412: FAM-CTGCAGAAGTGCCTGGCAGTG-TAMRA, FAM-CTGCAGAAGCGCCTGGCAGTG-TAMRA.

Statistical analysis

Data analysis was performed using SPSS statistical software version 20.0. Normally distributed data were analyzed using Student’s t-test and represented as mean ± SD and, whereas non-normally distributed measures are expressed as M (P25, P75). The Mann–Whitney U-test was used for comparison between two groups. The chi-square test was used to analyze categorical variables, which are presented as percentages. One-way analysis of variance was used to analyze the differences between ApoE alleles and blood lipids. Logistic regression analysis was performed to evaluate the interactions between ApoE polymorphisms and various factors (age, sex, and prevalence of hypertension) in T2DM. Statistical significance was set at p < 0.05. Statistical power analysis was performed using Power and Simple Size Calculation software (Dupont & Plummer, 1998).

Results

Population characteristics

The demographics and clinical traits of the participants in the current study are shown in Table 1. The levels of TC, HDL-C, LDL-C, Apo-A1, and Apo-B were statistically different (p < 0.05), but not in age, proportion of females, prevalence of hypertension, and TG levels between the T2DM group, which included 743 patients, and the control group, which included 995 individuals without T2DM. The power analysis indicates that our study population can provide sufficient statistical power (0.77) to test the association between SNP rs429358 and T2DM under the assumptions of a type I error of 0.05, a minor allele frequency of 0.129 (in East Asian according to 1000Genomes data), and an odds ratio (OR) of 1.3. However, power may not be sufficient for rs7412 (0.68), which has a minor allele frequency of 0.10.

Association of ApoE Genotypes and haplotypes frequencies with T2DM

Six ApoE genotypes were found in patients with T2DM, with the ɛ3/ɛ3 and ɛ3/ɛ4 types having the highest genotype frequencies (60.83% and 23.96%, respectively), followed by 10.77% with ɛ2/ɛ3, 2.15% with ɛ4/ɛ4, 1.75% with ɛ2/ ɛ4 (1.75%), and 0.54% with ɛ2/ɛ2. The frequencies of the ɛ2, ɛ3, and ɛ4 alleles were 6.80%, 78.20%, and 15.01%, respectively. Genotypic association revealed that the prevalence of the genotype ɛ3/ɛ4 was significantly higher in T2DM cases (23.96%) than that in controls (18.19%) (χ2 = 8.629, p = 0.003), and the frequency of the allele ɛ4 was found to be substantially associated with T2DM (15.01% in T2DM cases and 11.96% in controls; χ2 = 6.866, p = 0.009). There were no significant associations between other ApoE genotypes (ɛ2/ɛ2, ɛ2/ɛ3, ɛ2/ɛ4, ɛ3/ɛ3, and ɛ4/ɛ4) or alleles (ɛ2 and ɛ3) and T2DM risk (Table 2).

Association analysis for the ApoE alleles with lipid profiles in patients with T2DM

We additionally examined the association between ApoE alleles (ɛ2, ɛ3, and ɛ4) and serum lipid levels in patients with T2DM in the current population because ApoE polymorphisms have been linked to lipid profiles. We excluded individuals with the ApoE ɛ2/ɛ4 genotype because ɛ2 and ɛ4 alleles play opposing roles in lipid metabolism. We combined individuals with ɛ2 carriers (ɛ2/ ɛ2, ɛ2/ ɛ3) into the E2 group and ɛ4 allele carriers (ɛ3/ɛ4, ɛ4/ɛ4) into the E4 group, and individuals with ɛ3/ ɛ3 were considered as the third E3 group. The results showed that among patients with T2DM, participants in the E4 (ɛ3/ɛ4, ɛ4/ɛ4) group had significantly higher TG, TC, LDL-C, and ApoE-A1 levels than those in the E3 (ɛ3/ɛ3) group, while participants in the E2 group (ɛ2/ɛ2, ɛ2/ɛ3) showed lower TC, LDL-C, and ApoE-A1 levels than those in the E3 (ɛ3/ɛ3) group. No significant differences were found among the groups (Fig. 1).

Table 1 Comparison of the demographic and clinical characteristics of T2DM patients and non- T2DM controls.

	T2DM (n = 743)	Non-T2DM (n = 995)	χ2/z	p values*	
Age, [M (P25,P75), years]	69(59, 79)	69(56, 84)	−0.104	0.917	
Female, n (%)	393(52.89%)	543(54.57%)	0.483	0.487	
Hypertension, n (%)	337(45.35%)	430(43.21%)	0.791	0.374	
TG, [M (P25,P75), mmol/L]	1.18(0.86, 1.73)	1.21(0.87, 1.77)	−0.677	0.499	
TC, [M(P25,P75), mmol/L]	4.54(3.65, 5.35)	4.25(3.55, 5.02)	−3.781	<0.001	
HDL- C, [M (P25,P75), mmol/L]	1.04(0.85, 1.32)	1.07(0.87, 1.28)	−3.557	<0.001	
LDL- C, [M (P25,P75), mmol/L]	2.48(1.771, 3.17)	2.44(1.86, 3.09)	−2.752	0.006	
Apo- A1, [M (P25,P75), g/L]	1.01(0.85, 1.21)	1.09(0.93, 1.29)	−2.832	0.005	
Apo- B, [M (P25,P75), g/L]	0.83(0.61, 1.16)	0.81(0.62, 1.04)	−2.117	0.034	
Notes.

* Normal distribution date are represented by mean ± SD and analyzed using Student’s t-test, Non-normally distributed measures are expressed as M (P25, P75) and the Mann–Whitney U-test was used for comparison between two groups. The Chi- square test was used for analyzing categorical variables, which were presented as percentages.

Table 2 Genotypes and allele distribution of APOE gene in T2DM patients and control participants.

Genotypes	ɛ2/ɛ2	ɛ2/ɛ3	ɛ2/ɛ4	ɛ3/ɛ3	ɛ3/ɛ4	ɛ4/ɛ4	
T2DM	4
(0.54%)	80
(10.77%)	13
(1.75%)	452
(60.83%)	178
(23.96%)	16
(2.15%)	
Non-T2DM	5
(0.50%)	123
(12.36%)	21
(2.11%)	647
(65.03%)	181
(18.19%)	18
(1.81%)	
χ2	0.011	1.049	0.289	3.213	8.629	0.263	
p Values	0.918	0.306	0.591	0.073	0.003	0.608	
Alleles	ɛ2	ɛ3	ɛ4				
T2DM	101
(6.80%)	1162
(78.20%)	223
(15.01%)				
Non-T2DM	154
(7.74%)	1598
(80.30%)	238
(11.96%)				
χ 2	1.11	2.305	6.866				
p Values	0.292	0.129	0.009				

Figure 1 Association among serum lipid levels and APOE alleles in T2DM patients.

Note: p value shows the differences compared between groups (E2, E3, and E4). * p < 0.05** p < 0.01, *** p < 0.001, **** p < 0.0001.

Logistic regression analysis of T2DM in the central Chinese Han population

Logistic regression analysis was performed with T2DM as the dependent variable and blood lipid concentration and ApoE allele frequency as the independent variables. After adjusting for potential covariates, such as sex, age, and hypertension, individuals with high levels of TC (OR 1.21, 95% CI [1.1–1.32], p < 0.001), LDL-C (OR 1.29, 95% CI [1.17–1.41], p < 0.001), Apo-B (OR 1.90, 95% CI [1.31–2.85], p =0.001), and ApoE ɛ4 allele(OR 1.41, 95% CI [1.12–1.78], p = 0.004) had significantly high risks of T2DM. Therefore, serum TC, LDL-C, Apo-B levels, and ApoE ɛ4 allele were considered independent risk factors for T2DM (Fig. 2).

Figure 2 Logistic regression analysis of risks of T2DM in the central Chinese Han population.

Discussion

T2DM is a major contributor to the substantial global economic burden and one of the leading cause of mortality and morbidity worldwide (Khan et al., 2020). Recent clinical and epidemiological investigations have focused on the association between genetic susceptibility and gene polymorphisms in T2DM (Gloyn & Drucker, 2018). Because insulin resistance affects enzymes involved in lipid metabolism, T2DM is typically linked to dyslipidemia (Athyros et al., 2018). Decreased HDL-C and elevated TG and LDL-C levels are characteristics of diabetic dyslipidemia (Lemos, Torrinhas & Waitzberg, 2023). Understanding the genetic predisposition to diabetes and lipid profiles will enable us to identify high-risk populations, which will enable early detection and prevention, specialized medication therapy, and gene therapy. This will also help us to unveil the molecular pathogenesis of diabetes and dyslipidemia. ApoE is an important apolipoprotein involved in lipoprotein metabolism (Dominiczak & Caslake, 2011). Different isoforms of this polymorphic protein are linked with changes in lipid and lipoprotein levels, which, in turn, are associated with diabetes and cardiovascular risk (Marais, 2019). In this study, we examined the relationship between T2DM and ApoE gene polymorphisms and its effects on plasma lipid markers.

In the present study, genotypes ɛ3/ɛ3 and ɛ3/ɛ4 were the most common alleles of ApoE in the central Chinese Han population, accounting for 60.83% and 23.96%, respectively, and are similar to those of another study in the Chinese Han population (Long et al., 2019). Furthermore, the major findings of this study were that genotypes ɛ3/ɛ4 and allele ɛ4 conferred a risk for T2DM. However, alleles ɛ2 and genotypes ɛ2/ɛ2, ɛ2/ɛ3, ɛ2/ɛ4, and ɛ4/ɛ4 were not associated with T2DM risk. Our findings show that ApoE polymorphisms may be involved in the regulation of overall metabolic abnormalities. Our study is the first to reveal ApoE genotypes related with lipids in patients with diabetes in an older Chinese community. However, there are some differences between our study and the Korean population, where none of the ApoE polymorphisms, alleles (genotype), or haplotypes were significantly associated with T2DM risk, but only with lipids (Seo et al., 2021). This variation may be explained by genetic variations between populations or it may be related to our sample’s slightly larger size (743 patients with T2DM and 995 unaffected controls) compared to the Korean study population (352 patients with T2DM and 1,084 unaffected controls). Finally, logistic regression analysis showed that allele ɛ4 was associated with a significantly high T2DM risk (OR 1.41, 95% CI [1.12–1.78]; p = 0.004). Another study also proposed that compared with controls, the frequencies of genotype ɛ3/ ɛ4 (20.8% in T2DM cases versus 11.7% in controls, p = 0.04) and allele ɛ4 (14.3% in T2DM cases versus 8.3% in controls, p = 0.03) in patients with T2DM were significantly higher, which indicated that ɛ4 alleles increased T2DM risk by 1.64-fold, which is consistent with our study (Liu et al., 2019).

The TOMM40- ApoE region was found to be positively related to T2DM in a recent multi-ethnic Genome-wide association study (GWAS) (Cook & Morris, 2016), and this association was effectively replicated in various cohorts, including Asians (Long et al., 2019; El-Lebedy, Raslan & Mohammed, 2016). Consistent with the current study, the genetic influences of ApoE polymorphisms and genotype on T2DM susceptibility and lipid profiles, which have been attributed to T2DM, were determined in the Korean population (Seo et al., 2021). We searched the public DIAGRAM type 2 diabetes GWAS data and found that the SNP in the ApoE locus was significantly associated with T2DM in European ancestry, with rs429358 as the leading SNP (p = 1.4 ×10−10, beta = 0.12, number of T2DM cases = 26,676, number of controls = 132,532) (Fig. 3) (Morris et al., 2012). For rs7412, it was significantly associated with high cholesterol (P = 1.729×10−222, number of T2DM cases = 55,265, number of controls = 396,999, from UK biobank data) (Canela-Xandri, Rawlik & Tenesa, 2018), total cholesterol levels (p = 1.56×10−283, number of participants = 187,365), concentration of small LDL particles (p = 1.44×10−59, number of participants = 19,273), and concentration of medium LDL particles (p = 1.65×10−89, number of participants = 19,273) (Kettunen et al., 2016), although it was not significantly associated with T2DM risk in the DIAGRAM type 2 diabetes consortium (p = 0.21) (Morris et al., 2012). All these GWAS results and the genetic association in other independent populations demonstrated that the results of our study were robust.

Figure 3 The association between rs429358 and T2DM in European ancestry according to DIAGRAM type 2 diabetes GWAS database.

The GWAS summary statistics data of type 2 diabetes was downloaded from http://diagram-consortium.org, and the plots were generated by Locuszoom (https://statgen.sph.umich.edu/locuszoom/).

According to several studies, both the ApoE ɛ 2 and ɛ4 alleles have been linked to T2DM, and the ApoE ɛ4 allele is an independent risk factor for T2DM and cardiovascular disease, including coronary artery disease (Alagarsamy, Jaeschke & Hui, 2022a; Eichner et al., 2002). Although ApoE genetic variations, particularly in the ɛ4 allele, have been linked to T2DM susceptibility in a number of populations, including the Chinese and Korean populations in the current study, other Thai, Chilean, Indian, Mumbai, and Rancho Bernardo Studies found no association between ApoE polymorphism and T2DM (Jia et al., 2019; Oh & Barrett-Connor, 2001; Srirojnopkun et al., 2018; Pitchika et al., 2022). The ɛ2 allele may be a risk factor for diabetes according to a study conducted on Brazilian patients (Errera et al., 2006). These discrepancies may be due to racial and geographical differences. The two ApoE polymorphisms, rs429358 and rs7412, are located in the coding region of exon 4, which overlaps with a clearly defined CpG island, distinguishing the three prevalent ApoE alleles, ɛ2, ɛ3, and ɛ4 (Bezuch et al., 2021). Both SNPs alter the number of CpG dinucleotides, which are the main locations for DNA methylation, in addition to the protein codon. Foraker et al. (2015) demonstrated that the ApoE CpG island’s DNA methylation landscape is altered by the presence of the ɛ4 allele in brain tissue, and this epigenetic change increases the risk of Alzheimer’s disease. In a separate study, Ereqat et al. (2022) discovered that SNPs (rs769446, rs449647, and rs405509) in CpG sites in the ApoE promoter were associated with the level of DNA methylation in peripheral blood cells and that there was a difference in DNA methylation at these CpGs between patients with diabetic dyslipidemia and controls. However, it is unclear whether methylation of the ApoE gene affects T2DM, possibly because of the tissue-specific nature of methylation.

Overall, various factors, including age, sex, ethnicity, alcohol use, dietary fat, drug therapy, gene-gene and gene-environment interactions, pathogenesis of diabetic dyslipidemia, various complications of diabetes in the study participants, and the number of participants, can explain the inconsistent findings among ApoE polymorphisms and T2DM and serum lipids.

ApoE polymorphisms affect metabolic disorders, dyslipidemia, and T2DM via multiple processes (Alagarsamy, Jaeschke & Hui, 2022). Several ApoE polymorphisms have been linked to impaired glucose metabolism and a high risk of T2DM (Higuchi, Izquierdo & Haeusler, 2018). It has been proposed that the effects of ApoE on inflammation, beta cell function, and insulin resistance may mediate these effects (He et al., 2020). The ApoE ɛ2 allele has been associated with a decreased likelihood of developing diabetes; however, the ApoE ɛ4 allele has specifically been linked to impaired glucose tolerance and insulin resistance. Although the precise mechanisms underlying these correlations are still unclear, they may be related to inflammation, altered lipid metabolism, and adipose tissue malfunction. Further studies are required to completely understand the function of ApoE polymorphisms in the pathogenesis of diabetes and to consider possible therapeutic approaches that target this pathway.

Our study showed that Apo-B levels are an independent risk factor for T2DM in the elderly Chinese population. Apo-B is a protein present on the surface of some lipoproteins, such as VLDL and LDL, which contain high levels of cholesterol (Verges, 2022). The pancreas and other distal organs, such as the liver, receive lipids (cholesterol and triglycerides) via Apo-B (Ying et al., 2021). There data support the idea that T2DM may be highly likely in people with elevated levels of Apo-B (Karagiannidis et al., 2022). The accumulation of fatty deposits in the pancreas, which can hinder insulin secretion and increase the risk of diabetes, has been linked to increased blood levels of Apo-B-containing lipoproteins (Richardson et al., 2021). Furthermore, studies have shown that Apo-B is a more accurate indicator of diabetes risk than conventional lipid measurements, such as LDL and HDL cholesterol (Lamantia, Sniderman & Faraj, 2016; Gao et al., 2021a; Gao et al., 2021b). This indicates that Apo-B may be a key player in the onset and progression of diabetes and may serve as a valuable biomarker for identifying people at high risk of contracting the disease.

One limitation of the current study is the sample size; our relatively modest sample size may result in inadequate power and cause false positives when evaluating the association between rs7412 and T2DM. Therefore, the association between rs7412 and T2DM requires further validation in a relatively larger ethnic races or populations.

In the current investigation, among T2DM participants, ɛ4 carriers were linked to greater levels of TG, TC, LDL-C, ApoE-A1, and Apo-B than ɛ3 carriers, whereas ɛ2 carriers were linked to lower levels of TC, LDL-C, and ApoE-A1 than ɛ4 carriers. The primary explanation for this is that the LDL receptor has a high affinity for the ɛ4 isomer. Compared to ɛ3, ɛ4 increases the affinity for the LDL receptor and accelerates lipoprotein uptake in the liver, increasing the content of free cholesterol in liver cells. This feedback regulation then downregulates the expression of LDL receptors on the surface of liver cells, reducing the uptake of LDL by the liver, decreasing the rate of cholesterol clearance in the blood, and increasing TC and LDL-C levels accordingly.

Conclusions

The current study indicated that the genotype ɛ3/ɛ4 and allele ɛ4 of ApoE were associated with T2DM risk in the Chinese Han population in central China. Moreover, in patients with T2DM, participants in the E4 (ɛ3/ɛ4, ɛ4/ɛ4) group had significantly higher lipid profiles than those in the E3 (ɛ3/ɛ3) group, while participants in the E2 group (ɛ2/ɛ2, ɛ2/ɛ3) showed lower TC, LDL-C, and ApoE-A1 levels than those in the E3 (ɛ3/ɛ3) group. Therefore, the results of the current study may help in understanding ApoE polymorphisms and lipid profiles in a Chinese Han population.

Supplemental Information

Supplemental Information 1 Raw data

Click here for additional data file.

Additional Information and Declarations

Competing Interests

Author Contributions

Data Availability

The authors declare there are no competing interests.

Yali Zeng performed the experiments, prepared figures and/or tables, authored or reviewed drafts of the article, and approved the final draft.

Shuang Wen performed the experiments, analyzed the data, prepared figures and/or tables, and approved the final draft.

Lijun Huan analyzed the data, prepared figures and/or tables, and approved the final draft.

Liang Xiong analyzed the data, prepared figures and/or tables, and approved the final draft.

Botao Zhong conceived and designed the experiments, authored or reviewed drafts of the article, and approved the final draft.

Pengyun Wang conceived and designed the experiments, authored or reviewed drafts of the article, and approved the final draft.

The following information was supplied regarding data availability:

The raw data is available in the Supplemental File.

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
