# Peer review of "Association of ApoE gene polymorphisms with serum lipid levels and the risk of type 2 diabetes mellitus in the Chinese Han population of central China"

_PeerJ, doi:10.7717/peerj.15226_

## Round 0.1 · original submission · Major Revisions

As you will see both reviews have raised some comments for you to address. It is important that you address these comments fully and completely and I would strongly urge to carefully address the English language used throughout the manuscript. While I appreciate the difficulties this may present the quality of language use is presently detracting from a potentially interesting story.

I have to say I was in two minds about whether to invite re-submission of this manuscript. I am concerned that genetic associations in a study of this relatively small size maybe false positives. Hence I think without replication the findings are perhaps not robust. I think as a minimum you need to do two things : first please discuss the power calculations and the statistical power of your study. Secondly comparisons with public with existing publicly available data is in my opinion also essential. For example the DIAGRAM type 2 diabetes consortium Make the data publicly available and this includes a matter analysis of East Asian diabetes.

Both the statistical question and comparisons with other data sets are in my opinion essential. Without this I'm afraid I will need to reject the submission. However I hope that you will consider carefully my suggestions together with those of the two anonymous reviewers of your manuscript and revise your study accordingly. If you need extra time to achieve this level of additional work please do consult with the editorial office who I am sure would be sympathetic.

·

Basic reporting

Association of ApoE gene polymorphisms with serum lipid levels and the risk of type 2 Diabetes mellitus in the Chinese Han population of central China

The concept of this work is comprehensive and well-established. However, proper language editing is required. Raw data was shared. Background and references are OK.

Experimental design

The design of work is well-constructed, but various reports addressed this idea in different ethnic subjects.
The details of primers used in this analysis are required.

Validity of the findings

The novelty of the work is not clear and must be addressed in details.
What is the pathogenesis of APOE polymorphisms with diabetes mellitus.

Reviewer 2 ·

Basic reporting

No comments

Experimental design

No comments

Validity of the findings

No comments

Additional comments

RE: Association of ApoE gene polymorphisms with serum lipid levels and the risk of type 2 Diabetes mellitus in the Chinese Han population of central China by Zeng et al
The study investigated the association between ApoE alleles with T2DM and blood lipids in Chinese Han population and provide valuable information to be compared with other ethnic groups. The manuscript is clearly written but the following points should be addressed:

1-Selection criteria for control group was not clear, Pleas clarify
2-The authors showed that Apo-B levels was also considered to be independent a risk factor for T2DM, can they explain the role of Apo-B and add some studies in the discussion section
3-Line 172: genotype [3/[3 and [3/[4 are the most common alleles: these are genotype not alleles: please correct
4-Lines 173-174: for 60.83% and 23.96% and are 174 similar to those of the other studies in Chinese Han population: please add the reference
5-Line 196-200: the authors should explain about the role of APOE methylation in the inconsistent findings between ApoE polymorphisms and T2DM (cite previous studies)
6-APOE should be in italics ,( for example line 152, figure 1 and table 2) please check the whole text!

---

## Round 0.2 · accepted · Accept

Thank you for your careful and considered response to the issues raised by the first round of review of your paper. I am satisfied that all points have been addressed and/or discussed and the statistical concerns I raised have been suitably signposted in the revision.

I am delighted the recommend this for acceptance now.